# Italian Validation of the Pittsburgh Sleep Quality Index (PSQI) in a Population of Healthy Children: A Cross Sectional Study

**DOI:** 10.3390/ijerph19159132

**Published:** 2022-07-26

**Authors:** Alessia Scialpi, Ester Mignolli, Corrado De Vito, Anna Berardi, Marco Tofani, Donatella Valente, Giovanni Galeoto

**Affiliations:** 1Department of Anatomical, Histological, Forensic and Orthopaedic Sciences, Sapienza University of Rome, 00185 Rome, Italy; scialpialessia@gmail.com (A.S.); estermignolli@gmail.com (E.M.); 2Department of Public Health and Infection Disease, Sapienza University of Rome, 00185 Rome, Italy; corrado.devito@uniroma1.it; 3Department Human Neurosciences, Sapienza University of Rome, 00185 Rome, Italy; anna.berardi@uniroma1.it (A.B.); marco.tofani@uniroma1.it (M.T.); donatella.valente@uniroma1.it (D.V.); 4IRCCS Neuromed, 86077 Pozzilli, Italy

**Keywords:** sleep, validation, reliability, children

## Abstract

Background: Sleep disorders are one of the most discussed topics in scientific literature every year. Although they are one of the most studied topics, in both adults and children, knowledge of sleep disorders and their treatment is still not completely clear, and there is a need to deepen and analyze these disorders on a country-by-country basis. However, research in the Italian literature reveals a scarce quantity of tools to evaluate sleep quality in children. The Pittsburgh Sleep Quality Index (PSQI) is probably the most commonly used retrospective self-assessment questionnaire in the adult population. Purpose: We aimed to validate and analyze the psychometric characteristics of this tool in order to detect and explore the presence of sleep disorders in a healthy Italian population of children throughout the country. Methods: Individuals aged between 3 and 16 years without symptoms of insomnia were included in this study. The reliability and construct validity of the PSQI were assessed according to Consensus-Based Standards for the Selection of Health Measurement Instruments (COSMIN) guidelines. Results: We enrolled 222 individuals in this study (mean age 11 years). The PSQI demonstrated good internal consistency (Cronbach’s α = 0.719). Test–retest reliability was assessed on a randomized subgroup of the sample (*n* = 35). The PSQI showed good test–retest reliability with an intraclass correlation coefficient of 0.829 for the total score (95% confidence interval: 0.662–0.914). The Pearson correlation coefficient, used for construct validity, showed a statistically significant positive correlation with the Sleep Disturbance Scale for Children (SDSC). Conclusion: The PSQI proved to be a very reliable and valid tool to investigate sleep experiences in children.

## 1. Introduction

Sleep is a physiological process during which a reduction in consciousness and metabolism occurs [1]. Sleep quality is a vital indicator of overall health and wellbeing. After 2 years of research, the National Sleep Foundation revisited the sleep needs guideline criteria for each age group. As a result, the amount of sleep recommended for school-aged children (6–13 years) was extended by 1 h to 9–11 h and the amount of sleep for adolescents (14–17 years) was extended by 1 h to 8–10 h. Regarding adults, young adults (18–25 years) need 7–9 h, adults (26–64 years) need 7–9 h, and the elderly (>65 years) need 7–8 h. As individuals grow older, the need for sleep decreases, with preschoolers thus requiring more hours of sleep than adults [2].

Sleep is a fundamental element for the wellbeing and healthy development of children. Between 0 and 3 months, an infant’s sleep has completely different characteristics than adult sleep as it consists of several phases that follow one another, and there is a greater presence of the light sleep phase (rapid eye movement (REM) phase) compared to the deep sleep phase (NREM phase). Infants sleep about 15 h splits between day and night; however, since sleep is predominantly in the REM phase, they are likely to wake frequently. From 4 months of age, infant sleep takes on characteristics that are increasingly similar to adult sleep as the development of the nervous system allows them to sleep for up to eight consecutive hours without the need for parental intervention. During this period, the child sleeps about 8–10 h during the night and 4–5 h during the day. From 8 to 12 months, the amount of sleep decreases, and there is a reduction in restless sleep where phases of light sleep and deep sleep alternate several times during the night [3].

Poor-quality sleep can have negative repercussions in children, including cognitive, motor, and behavioral consequences, as well as in terms of participation in all activities of daily life. Some authors have pointed out that about 37% of children between 4 and 10 years have sleep disorders that often become chronic [4]. It was also noted that children with physical, psychiatric, or other disabilities have a greater likelihood of developing these disorders [4,5]. 

The sleep disorders most frequently observed in children are insomnia, sleep breathing disorders, circadian rhythm disorders, and parasomnia [5]. According to the International Classification of Sleep Disorders, third edition (ICSD-3) and the Diagnostic and Statistical Manual of Mental Disorders, fifth edition (DSM-V), a diagnosis of insomnia is mainly based on the evaluation of self-reported symptoms, and the sleep diary has become a standard tool to assess the patient’s self-reported insomnia perception. In clinical practice, the use of structured and semi-structured clinical interviews to assess sleep is increasing [2].

Several questionnaires have been developed to obtain numerical values that can be scientifically analyzed for childhood sleep disorders. The Children’s Sleep Habits Questionnaire (CSHQ) [6] is a questionnaire consisting of 45 items that is completed by parents with the aim of investigating the frequency with which their child experienced sleep disturbances during the previous month.

The Pediatric Sleep Questionnaire (PSQ) is one of the most widely used sleep quality assessment questionnaires for children. It is divided into 10 sections with a total of 22 items and can be easily completed by the parents of children aged between 2 and 18 years [7].

The Sleep Disturbance Scale for Children (SDSC) [8] is a scale that evaluates the frequency of sleep disturbances in the 6 months prior to its administration. The main strengths of the SDSC are the simplicity of the questions and the consequent immediacy in measuring the quality of sleep.

The Pittsburgh Sleep Quality Index (PSQI) [9] is probably the most commonly used retrospective self-assessment questionnaire. This questionnaire measures sleep quality in the previous month.

A systematic review (2016) [10] demonstrated that the scale has excellent psychometric properties for all languages and for all pathologies studied. Internal consistency was reported in 12 studies [11,12,13,14,15,16,17,18,19,20,21,22]; in almost all studies, Cronbach’s alpha values met the cutoff point for a positive rating for within- and between-group comparisons, ranging from 0.70 to 0.83. The test–retest reliability of the PSQI was evaluated in three studies with different coefficient analysis [17,18,23]. A strong association (construct validity) was uncovered between the PSQI total score and the insomnia severity index (ISI) total score, sleep problems from symptom experience reports, short form-36 health survey vitality score, sleep restlessness score, and sleep efficiency score from the sleep diary. A moderate association (Pearson’s correlation coefficient 0.5) was found between the PSQI and disability scores, depression, tension/anxiety, and confusion.

On the other hand, validations in the pediatric population are fewer with the first validation in these samples being performed in 2017. Two studies, conducted in Australia [24] and Brazil [25], evaluated psychometric properties in a sample of secondary-school adolescents. A recent paper evaluated the psychometric properties of the Chinese version in a sample of childhood cancer survivors [26]. Lastly, a study was conducted in Canada on a population of children with chronic pain [27].

The objective of this cross-sectional study was to evaluate the psychometric properties of the PSQI in a pediatric population by analyzing the internal consistency, reliability, and construct validity. For construct validity, the hypothesis was to find a positive statistically significant correlation with SDSC.

## 2. Materials and Methods

This study was conducted by a research group from Sapienza University of Rome experienced in validating outcome measures [28,29,30]. Following Consensus-Based Standards for the Selection of Health Measurement Instruments (COSMIN) [31] guidelines, the validation of the scale was performed on a sample of healthy Italian children. A sample of subjects was recruited and administered the DSM-V criteria for insomnia disorder. The PSQI was administered only to subjects who did not meet the diagnostic criteria. The construct validity of the PSQI was obtained by comparing the results of the scale with those of an already validated scale, i.e., the SDSC, which was administered concurrently with the PSQI. To evaluate test–retest reliability, a subgroup of participants was subjected to a second administration of the PSQI alone after about 3 days.

### 2.1. Participants

Only individuals who met the following inclusion criteria were included in the study: (1) aged between 3 and 16 years; (2) absence of insomnia symptoms according to DSM-V diagnostic criteria; (3) ability to communicate in Italian; (4) signature of the parent or guardian providing informed consent to the processing of personal data.

Participants interested in taking part in the study were informed [32,33] about the purpose and methods of the study and were registered. Participants were recruited from schools in different parts of Italy.

### 2.2. Assessment Tools

Pittsburgh Sleep Quality Index: The 19-item PSQI is the most used retrospective self-report questionnaire, which measures sleep quality over the previous month [18]. Seven clinically derived domains of sleep difficulties (sleep quality, sleep latency, sleep duration, habitual sleep efficiency, sleep disturbances, use of sleeping medications, and daytime dysfunction) are assessed by the questionnaire. Taken together, these sleep domains are scored as a single factor of global sleep quality. Usually, a global score higher than 5 is considered as an indicator of relevant sleep disturbances in at least two components or of moderate difficulties in more than three components [18]. The scale was validated in the Italian language in 2013 [9].

Sleep Disturbance Scale for Children: The SDSC assesses disorders of initiating and maintaining sleep, sleep breathing disorders, disorders of arousal, sleep/wake transition disorders, disorders of excessive somnolence, and sleep hyperhidrosis. The six subscales are scored on a five-point Likert scale evaluating the past 6 months, and this tool has been translated into several languages with adequate results with respect to validity and reliability. The scale was validated in the Italian language in 2013 [34].

### 2.3. Reliability and Validity

All statistical analyses were performed with Statistical Package for Social Sciences (SPSS) software(Armonk, NY, USA). The reliability and validity of the PSQI were assessed according to COSMIN guidelines. Internal consistency was examined with Cronbach’s α in order to evaluate the interrelation of the items and the homogeneity of the scale. A value of α ≥ 0.7 is commonly considered an acceptable indicator of the satisfactory homogeneity of the items within the total scale [35]. In order to evaluate test–retest reliability, a subgroup of the population underwent a second administration. A timeframe of about 3 days was considered long enough for the participant to not remember previous responses given. The intraclass correlation coefficient (ICC) was calculated to determine the degree to which repeated measurements were free from measurement errors. A value equal to or greater than 0.70 is commonly considered optimal. To assess construct validity, the Italian version of the SDSC was administered to the entire population. The correlation between these scales was evaluated with Pearson’s correlation coefficient, with a value greater than 0.5 or less than −0.5 indicating an acceptably high level of correlation, with a positive value indicating a positive linear correlation and a negative value indicating a negative linear correlation. A *p*-value <0.05 was considered statistically significant.

## 3. Results

### 3.1. Participants

For the validation of the tool in the Italian population, 239 subjects were recruited from all over Italy. Of these, 222 (93%) met the inclusion criteria. The demographic characteristics of the population are summarized in Table 1.

### 3.2. Evaluation of the Psychometric Properties of the PSQI

The PSQI demonstrated good internal consistency (Cronbach’s α = 0.719). Evaluating Cronbach’s alpha when an item was deleted showed that all the components combined to structure a good internal consistency (Table 2). 

Test–retest reliability was assessed on a randomized subgroup of the sample (*n* = 35). The PSQI showed good test–retest reliability with an ICC for the total score of 0.829 (95% confidence interval: 0.662–0.914). The Pearson correlation coefficient for construct validity showed statistically significant values of a positive correlation between subscale and total scores of the scale, indicating an excellent correlation between PSQI and SDSC total scores, with a value of 0.767 (*p* < 0.01). This shows that the PSQI is capable of evaluating the same constructs as the SDSC. Results are shown in Table 3.

## 4. Discussion

Sleep disorders are one of the most discussed topics in scientific literature every year [36,37,38] due to the heterogeneous spectrum with which these disorders present themselves and the number of diseases and psychological conditions associated with them [39,40,41]. Although sleep disorders, in both adults and children, receive a lot of attention, knowledge regarding sleep disorders and their treatment is not completely clear, and there is a need to better investigate these disorders on a country-by-country basis. However, research in the Italian literature reveals a scarce quantity of tools to evaluate sleep quality in children. 

The PSQI is validated in many languages and has been studied in strictly clinical (*n* = 15 or nonclinical (*n* = 13) samples, or both (i.e., featuring nonclinical and clinical groups of participants within same study) (*n* = 9). The samples represented a variety of clinical disorders, including cancer, schizophrenia, and chronic fatigue syndrome, as well as healthy participants of varying ages, sexes, and races and ethnicities [10].

The PSQI is probably the most commonly used retrospective self-assessment questionnaire in adults. It was, therefore, decided to validate and analyze the psychometric characteristics of this tool to detect and explore the presence of sleep disorders in a sample of healthy Italian children throughout the country. 

The results of this study suggest that the PSQI version for children is a valid and reliable tool. The instrument demonstrated a good degree of internal consistency (Cronbach’s α = 0.719) and reliability, with an ICC of 0.829. Correlation with the SDSC showed that the PSQI has good construct validity. 

The PSQI has been validated in numerous languages in adult populations as stated in a 2016 systematic review which analyzed the validations of the scale in all languages in both healthy and pathological samples [10].

On the other hand, validations in the pediatric population are fewer. Two studies, conducted in Australia [24] and Brazil [25], evaluated psychometric properties in a sample of secondary school adolescents. The instrument obtained excellent psychometric properties with an internal consistency value of 0.73 (Cronbach’s Alpha) for the Australian study and 0.71 for the Brazilian study. 

A recent paper evaluated the psychometric properties of the Chinese version in a sample of childhood cancer survivors; the scale showed excellent data both of internal consistency with a Cronbach’s alpha value of 0.71 and an excellent reliability value with an ICC of 0.90 [26].

Lastly, a study conducted in Canada on a population of children with chronic pain showed that the scale is also valid and reliable for this population with a Cronbach’s alpha equal to 0.71 [27].

The Italian version of the PSQI, thus, proved to be a valid and reliable tool that is easy to use and able to quickly identify the presence of sleep disorders in healthy children. This scale investigates the child’s sleep in the month prior to its administration, thus identifying the possible need to intervene in order to counteract the presence of sleep disorders that adversely affect the daily life of the child. Future research should evaluate potential predictors of sleep quality by using diaries to document behaviors that might be related to sleep quality, such as the practice of sport. Data regarding these habits could also be included in structural equation modeling to provide an empirical validation of the relationship between these behaviors and sleep quality.

### Limitation 

Despite the excellent psychometric properties demonstrated, the study had some limitations. The sample size did not allow evaluating the existing differences with respect to the characteristics of the sample; as there are no other gold standards validated in Italian, the construct validity was not carried out with other scales widely used internationally.

## 5. Conclusions

The PSQI proved to be a very reliable and valid tool to investigate sleep experiences in the month prior to its administration. It can now be used by Italian professionals investigating sleep disturbances in research and clinical practice.

## Figures and Tables

**Table 1 ijerph-19-09132-t001:** Demographic characteristics of the included participants.

	Without Insomnia	With Insomnia
Mean age (standard deviation), years	10.4 (4.03)	11.5 (4.82)
Gender (%)		
Male	93 (40.26)	3 (1.30)
Female	129 (55.84)	6 (2.60)

**Table 2 ijerph-19-09132-t002:** Internal consistency: Cronbach’s alpha with item deleted.

	Mean When Item Was Deleted	Variance When Item Was Deleted	Item–Total Correlation	Cronbach’s Alpha When Item Was Deleted
Component 1	2.55	4.553	0.660	0.622
Component 2	2.39	4.666	0.401	0.709
Component 3	2.94	5.483	0.471	0.680
Component 4	2.91	5.940	0.223	0.732
Component 5	2.52	5.225	0.542	0.663
Component 6	3.10	6.073	0.288	0.716
Component 7	2.65	5.036	0.498	0.669

**Table 3 ijerph-19-09132-t003:** Construct validity: Pearson’s correlation coefficient between the Sleep Disturbance Scale for Children (SDSC) and the Pittsburgh Sleep Quality Index (PSQI).

SDSC	Component							Total Score
	1	2	3	4	5	6	7	
DIMS	0.642 **	0.659 **	0.494 **	0.203 **	0.499 **	0.300 **	0.465 **	0.783 **
SBD	0.202 **	0.291 **	0.045	0.167 *	0.253 **	0.037	0.153 *	0.286 **
DOA	0.303 **	0.108	0.230 **	0.190 **	0.391 **	0.104	0.227 **	0.351 **
SWTD	0.411 **	0.344 **	0.242 **	0.105	0.419 **	0.184 **	0.346 **	0.487 **
DOES	0.473 **	0.369 **	0.385 **	0.117	0.381 **	0.108	0.503 **	0.559 **
SH	0.187 **	0.131 *	0.065	0.083	0.272 **	0.341 **	0.208 **	0.284 **
G	0.627 **	0.561 **	0.440 **	0.208 **	0.571 **	0.287 **	0.530 **	0.767 **

* *p* < 0.05; ** *p* < 0.01; DIMS = disorders initiating and maintaining sleep; SBD = sleep breathing disorders; DOA = disorders of arousal; SWTD = sleep/wake transition disorders; DOES = disorders of excessive somnolence; SH = sleep hyperhidrosis; G = general factor.

## Data Availability

Data that support the findings of this study are available from the corresponding author upon reasonable request.

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
