# Peer review of "Italian Validation of the Pittsburgh Sleep Quality Index (PSQI) in a Population of Healthy Children: A Cross Sectional Study"

_ijerph, 2022, doi:10.3390/ijerph19159132_

Round 1

Reviewer 1 Report

Abstract: word count 288!? – look in the instructions for authors

Introduction section:

Line 84-98 : should be part of Discussion section

“For construct validity the hypothesis was to find positive statisti cally significant with SDSC” – need to rewrite..sounds confusing

Materials and Methods

Line 111:  outcome measures.13–2223 – references?

Result section: too short

Discussion section: also too short

First paragraph sounds like repeating the introduction section

Need to expand the number of references and comment their results

Conclusion section:

“It can now be used by Italian professionals investigating sleep disturbances in research and clinical practice.” – this sounds tendentious

References: need to rewrite: look in the instructions for authors

Author Response

Date: July 5, 2022

Dear Editor,

We appreciate the opportunity to resubmit our article entitled “ITALIAN VALIDATION OF THE PITTSBURGH SLEEP QUALITY INDEX (PSQI) IN A POPULATION OF HEALTHY CHILDREN: A CROSS SECTIONAL STUDY.” We would like to thank the referees for the careful and constructive reviews. We have made corresponding changes directly to the manuscript where appropriate with changes tracked. The revised version of our manuscript accompanies this letter. All comments by the reviewer have been addressed. Based on his/her comments, we have made changes to the manuscript, which are detailed below.

Reviewer Comment

Response

Line #

Reviewer #1

Line 84-98 : should be part of Discussion section

84-49 was moved to discussion section

194-198

“For construct validity the hypothesis was to find positive statisti cally significant with SDSC” – need to rewrite..sounds confusing

The sentence has been rewrite

Line 111:  …outcome measures.13–2223 – references?

Sentence has been corrected

105, 106

Result section: too short

Authors did not add any text because they think that all the relevant information are present, more text has been added to discussion section

Discussion section: also too short

First paragraph sounds like repeating the introduction section

Need to expand the number of references and comment their results

All the results are commented (cronbach’s alpha, ICC and construct validity) and compared with previous studies. All the studies on PSQI have been cited

9-13

Conclusion section:

“It can now be used by Italian professionals investigating sleep disturbances in research and clinical practice.” – this sounds tendentious

The sentence has been mitigated

References: need to rewrite: look in the instructions for authors

References has been corrected

References section

We hope that the new version of our manuscript is acceptable for publication.

Best regards,

Reviewer 2 Report

The core of the manuscript, "Italian validation of the Piisburgh Sleep Quality Index in a population of healthy children" is valuable, but its presentation fails to come up to expectations

Introduction

Several references are lacking                                                                       Page 2, paragraph 2, line 57. Some authors ?                                                  paragraph 2, line 58: it was also noted ?                                                     paragraph 4, line 70-71. The children's Sleep habits Questionnaire ?        paragrah 8, line 90: Internal consistency was reported in 12 studies? paragraph 8, line 93: the test-retest reliability of the PSQI was evaluated in three studies?

Some assertions are vague                                                                             page 2, paragraph 8, line 90: in almost all studies                                      paragraph 8, line 97: a moderate association

Inaccurate                                                                                                         light sleep phase (REM phase)                                                                        deep sleep phase (NREM phase)        

 Unclear                                                                                                          line 84: The PSQI is validated in more language

The objective of this cross-sectional study is clearly expressed: to evaluate the psychometric properties of the PSQI in a pediatric population by analyzing the internal consistency, reliability and construct validity

2. Materials and methods

Concurrent validity and test-retest validity. Why no keeping the same wording throughout the study?

2.1. Participants

Clearly defined

2.2. Assessment tools

PSQI                                                                                                                SDSC                                                                                                             CSHQ ?                                                                                                           PSQ ?

Reliability and validity are not assessment tools. Prefer "Statistics"

3 Results                                                                                                         3.1. Participants. Is figure 1 necessary?

3.2. Reliability and validity                                                                            Prefer: evaluation of the psychometric properties of the PSQI

4. Discussion

The first paragraph is redundant with the introduction

Line 205: concurrent validity or construct validity?

References

Please homogenize the  formulation of references                                         ,  2. Hirshkowitz M et al                                                                                       3. Philbrook, L.E., Becker L.E. & Linde J                                                             12. Larche C.L.,Plnte I., Roy M., Ingelmo P.M. & Ferland C.E.

Author Response

Date: July 5, 2022

Dear Editor,

We appreciate the opportunity to resubmit our article entitled “ITALIAN VALIDATION OF THE PITTSBURGH SLEEP QUALITY INDEX (PSQI) IN A POPULATION OF HEALTHY CHILDREN: A CROSS SECTIONAL STUDY.” We would like to thank the referees for the careful and constructive reviews. We have made corresponding changes directly to the manuscript where appropriate with changes tracked. The revised version of our manuscript accompanies this letter. All comments by the reviewer have been addressed. Based on his/her comments, we have made changes to the manuscript, which are detailed below.

Reviewer Comment

Response

Line #

Reviewer #2

Several references are lacking Page 2, paragraph 2, line 57. Some authors? paragraph 2, line 58: it was also noted? paragraph 4, line 70-71. The children's Sleep habits Questionnaire? Paragraph 8, line 90: Internal consistency was reported in 12 studies? paragraph 8, line 93: the test-retest reliability of the PSQI was evaluated in three studies?

All references have been added

58, 60, 71, 86, 89

Some assertions are vague page 2, paragraph 8, line 90: in almost all studies paragraph 8, line 97: a moderate association

Sentence corrected

93

Inaccurate light sleep phase (REM phase) deep sleep phase (NREM phase)

NREM has been corrected

47

Unclear line 84: The PSQI is validated in more language

Sentence has been corrected

190

Concurrent validity and test-retest validity. Why no keeping the same wording throughout the study?

The same wording has been made

Throughout the text

PSQI SDSC CSHQ? PSQ?

Reliability and validity are not assessment tools. Prefer "Statistics"

Subheading has been corrected

140

3.2. Reliability and validity Prefer: evaluation of the psychometric properties of the PSQI

Subheading has been corrected

164

The first paragraph is redundant with the introduction

First paragraph has been modified

182-188

Line 205: concurrent validity or construct validity?

This term has been corrected

Please homogenize the formulation of references, 2. Hirshkowitz M et al 3. Philbrook, L.E., Becker L.E. & Linde J 12. Larche C.L., Plnte I., Roy M., Ingelmo P.M. & Ferland C.E.

References has been corrected

References section

We hope that the new version of our manuscript is acceptable for publication.

Best regards,

Round 2

Reviewer 1 Report

Discussion section needs to be more expanded: it is too short.

Line 215-217 - delete. You already mentioned that in line 202-206 

Conclusion

"It can now be used more evidence by Italian professionals investigating sleep disturbances in research and clinical practice." - rewrite - sound confusing

Author Response

Date: July 22, 2022

Dear Editor,

We appreciate the opportunity to resubmit our article entitled “ITALIAN VALIDATION OF THE PITTSBURGH SLEEP QUALITY INDEX (PSQI) IN A POPULATION OF HEALTHY CHILDREN: A CROSS SECTIONAL STUDY.” We would like to thank the referees for the careful and constructive reviews. We have made corresponding changes directly to the manuscript where appropriate with changes tracked. The revised version of our manuscript accompanies this letter. All comments by the reviewer have been addressed. Based on his/her comments, we have made changes to the manuscript, which are detailed below.

Reviewer Comment

Response

Line #

Reviewer #1

Discussion section needs to be more expanded: it is too short.

223-227

Line 215-217 - delete. You already mentioned that in line 202-206

215-217 is not a repetitio it is the fourth study in the pediatric population:

[25] Australia secondary school adolescents

[26] Brazil

secondary school adolescents

[27] Chinese cancer survivors

[28] Canada chronic pain

Conclusion

"It can now be used more evidence by Italian professionals investigating sleep disturbances in research and clinical practice." - rewrite - sound confusing

This sentence has been corrected

236, 237

Reviewer #2

We hope that the new version of our manuscript is acceptable for publication.

Best regards,

Giovanni Galeoto
